# Assessing Microbial Activity and Rhizoremediation in Hydrocarbon and Heavy Metal-Impacted Soil

**DOI:** 10.3390/microorganisms13040848

**Published:** 2025-04-08

**Authors:** Robert Conlon, David N. Dowling, Kieran J. Germaine

**Affiliations:** EnviroCORE Research Centre, South East Technological University (SETU), Kilkenny Road, R93 V960 Carlow, Ireland; robert.conlon@setu.ie (R.C.); david.dowling@setu.ie (D.N.D.)

**Keywords:** rhizodegradation, phytoremediation, petroleum hydrocarbons, microbial communities, heavy metals, bioremediation

## Abstract

Rhizodegradation enhances pollutant degradation through plant–microbe interactions in the rhizosphere. Plant roots provide a colonisation surface and root exudates that promote microbial abundance and activity, facilitating organic pollutant breakdown via direct microbial degradation and co-metabolism. This study assessed the rhizodegradation of weathered petroleum hydrocarbons (PHCs) in heavy metal co-contaminated soil in a microcosm-scale pot trial. Treatments included *Sinapis alba*, *Lolium perenne*, a *L. perenne* + *Trifolium repens* mix, and *Cichorium intybus*, alongside a non-planted control. After 14 weeks, PHC concentrations were analysed via gas chromatography, and rhizosphere microbial communities were characterised through sequencing. *Sinapis alba* achieved the highest PHC degradation (68%), significantly exceeding the non-planted control (*p* < 0.05, Kruskal–Wallis test). Hydrocarbon-degrading bacteria, including *KCM-B-112*, *C1-B045*, *Hydrogenophaga*, unclassified *Saccharimonadales* sp., and *Pedobacter*, were enriched in the rhizosphere, with the uncultured clade mle1-27 potentially contributing indirectly. Metals analysis of plant tissues showed that mustard could accumulate copper more than lead and zinc, despite higher concentrations of zinc and lead in the soil. These results highlight the potential of *S. alba* for rhizoremediation in PHC–heavy metal co-contaminated soils.

## 1. Introduction

In the last two decades, there has been a paradigm shift in the recognition of soil as a non-renewable, valuable resource with critical ecosystem services, such as media for food and fibre production, water filtration, carbon sequestration, and a habitat for biodiversity. Healthy soils are necessary for the continued provision of these ecosystem services. Yet, global soil health continues to deteriorate due to contamination, becoming an increasing burden on both the environment and society. There are many different pollutants that contribute to this; those of greatest concern are petroleum hydrocarbons (PHCs), chlorinated hydrocarbons, pesticides, herbicides and heavy metals [1]. The petrochemical industry continues to grow due to a combination of population growth and demand for petroleum-derived products. This growth contributes to increasing petroleum hydrocarbon pollution, which occurs during production, transport, usage and disposal of the oil or waste products. Once released into the environment, PHCs may form slicks on the surface of water bodies or accumulate in soils and sediments [2,3]. The physical and chemical interactions of soil and petroleum hydrocarbons make this type of soil pollution particularly harmful to the environment and challenging to resolve.

PHCs cause harm to an ecosystem via their mutagenic, carcinogenic and lipophilic effects. The lipophilic/hydrophobic nature of PHCs hinders liquid transfer and air diffusion within contaminated soil, leading to alterations in soil permeability, moisture content, pH and nutrient availability. As a result, the population and diversity of living organisms in these soils are often negatively affected [1,4]. Oil contamination also disrupts water flow in soil by reducing hydraulic conductivity, altering pore structure and restricting the movement of moisture to plant roots. Studies have shown that PHC contamination can lower hydraulic conductivity by approximately 10% due to the coating of soil particles with PHCs, which traps pore spaces and impedes water infiltration and retention [2]. If a pristine ecosystem is contaminated with petroleum hydrocarbons, physical and chemical changes occur rapidly. The microbial community is greatly altered, resulting in reduced and less diverse populations, disrupting the ecosystem services they may provide. Plants are also affected, due to reduced access to water and nutrients, reduced access to light and build-up of toxic chemicals in plant tissues. These factors lead to impaired plant growth. When both a soil’s microbial community and plant life are impaired by significant PHC pollution, an ecosystem is less likely to support higher-order lifeforms [5,6].

The landscape of environmental remediation is going through a necessary period of transition. Thermal treatment and chemical oxidation are two established methods that efficiently remove organic contaminants from soil but, in the process, alter or destroy the soil itself. Phytoremediation is an alternative candidate to these established techniques and can most simply be defined as the use of plants to degrade or remove environmental contaminants. It is often paired with bioaugmentation (the introduction of specific pollutant-degrading bacteria) and is an ever-improving method of removing or degrading PHCs from polluted soil [7].

Phytoremediation is considered an attractive alternative to physical–chemical methods because it is cost-effective, ecologically beneficial, in situ, passive and is more aesthetically pleasing. It utilises the natural activities of plants and their associated microorganisms to degrade contaminants in an effected ecosystem [8,9,10]. Rhizoremediation is an approach to phytoremediation that requires the utilisation of plant–microbe interactions within the plant rhizosphere to achieve the degradation of organic contaminants. By using the naturally evolved interactions between plants and appropriate microorganisms, improved or complete mineralisation of contaminants can be attained [11]. Plants in the process of rhizoremediation mostly act indirectly in the degradation process. Many factors affect the efficacy of rhizoremediation, including the species of plant used, the number of TPH-degrading microbes, physical–chemical properties of pollutants and environmental conditions. For example, the presence of heavy metals in soil, especially in high concentrations, will limit the effectiveness of hydrocarbon degradation, due to toxic effects [12,13,14].

As things stand, phytoremediation and, by extension, rhizoremediation do not fit the criteria of a highly efficient and consistent approach to environmental clean-up. Researchers have been trying to address this by focusing on improving the soil conditions, bioavailability and accessibility of compounds and plant growth conditions [12,15]. Thijs et al. [15] suggested that a deeper understanding of the ecology of the “meta-organism”, i.e., the plant and its associated microorganisms, will be more successful at optimising phytoremediation. Other studies such as He et al. (2022) found that Lolium perenne could degrade PHCs from 34 to 60% depending on soil type, with the highest results pertaining to sandy soils [16]. They found that soil type and root exudates were crucial factors in determining both PHC breakdown and cadmium uptake efficiency, though Lolium perenne was not effective at the phytoextraction of cadmium. Previous phytoremediation research relating to mustard (Sinapis alba) focused on its utility in the phytoextraction of heavy metals. For example, Płociniczak et al. (2013) found that mustard was effective for the phytoextraction of zinc (926 mg/kg), cadmium (32 mg/kg) and copper (60 mg/kg), especially in the presence of Pseudomonas fluorescens MH15 [17]. Importantly, this study also found no evidence of phytotoxicity.

Our study involved a greenhouse-based pot trial to evaluate the efficacy of several plant species, along with a biostimulation control, in degrading petroleum hydrocarbons in sandy-clay soil co-contaminated with hydrocarbons and heavy metals. The plant species used were Sinapis alba (mustard), two high-biomass-yielding varieties of Lolium perenne (ryegrass), a commercial mixture of Lolium perenne with Trifolium repens (ryegrass and white clover) and Cichicorium intybus (chicory). From this point forward in the text, the plants will be referred to by their common names. At the conclusion of the treatment period, we assessed both the concentrations of petroleum hydrocarbons and the shifts in microbial communities. The main objectives were to identify differences in hydrocarbon degradation rates and microbial community structures among the treatments, explore the factors driving these variations (e.g., microbial activity or plant influence) and, additionally, to assess the potential of the selected plant species as viable candidates for remediating hydrocarbon-contaminated soil with heavy metal co-contaminants.

## 2. Materials and Methods

### 2.1. Soil Conditions

The soil used in this experiment was previously described in Curiel-Alegre et al. [18]. It is a sandy-loam soil (11.8% clay, 29.8% silt and 58.3% sand) obtained from an industrial site in Toledo, Spain, with bulk density 1.51 g cm^−3^, highest water retention capacity (HWRC) 25.33%, pH (1:5) 7.1, electrical conductivity (EC 1:5) 0.839 dS m^−1^, loss on ignition (LOI) 3.85%, oxidizable organic carbon 2.59%, total N 0.02%, nutrients (mg kg^−1^): NH_4_–N 2.99; NO_3_–N 0.14; PO_4_–P 0.10, lime content 35.08%, trace elements (mg kg^−1^): As 77.3; Cd 7.8; Cr 14.9; Cu 8.5; Ni 9.9; Pb 339.2; Zn 680.5, and total petroleum hydrocarbons (PHCs): 4051 mg kg^−1^. The elevated concentrations of Pb (339.2 mg/kg), Zn (680.5 mg/kg), As (77.3 mg/kg), and Cd (7.8 mg/kg) in the soil notably exceed both typical background levels and the European Union’s regulatory thresholds for these heavy metals, confirming the soil’s contamination status [19]. While humic substances are known to influence metal binding, specific humic content was not measured in this study, limiting the interpretation of metal bioavailability and mobility. This land is an industrial site, the contamination of which would have reduced the diversity and abundance of plants species.

### 2.2. Experimental Layout

The soil was prepared for the experiment by air drying followed by sifting using a 2 mm sieve. The experiment comprised a randomised pot trial utilising 6 treatments: 5 plant treatments and a non-planted control (biostimulation with no plants), each consisting of 5 replicates. Total organic carbon in the soil was 3.43% (34,300 mg/kg). Then, 5 g/kg fertiliser (24:5 N:P, as 14% ammonia, 11% nitrate) was added to all treatment pots to promote biostimulation and create a ratio of approximately 170:10:1 C:N:P. This ratio was selected as it is close to the global average ratios of C:N:P in soil and takes into account that nitrogen and phosphorous are the limiting nutrients for microbial growth in soil [20]. A clean layer of nutrient-rich garden soil approximately 2 cm deep was added to the top of every pot (including biostimulation control) to facilitate germination of the plant seeds.

At the end of the trial, this clean layer of soil could easily be removed without mixing with the contaminated soil, thus avoiding any potential dilution of the soil. We employed a combination of monocot and dicot species to evaluate how different root structures influence their phytoremediation potential. The plants used were mustard, chicory, two varieties of high-yield ryegrass and a blend of ryegrass and white clover. Mustard and chicory are dicots chosen for their deep rooting capacity. Mustard and chicory have been used previously in phytoextraction of heavy metals [17,21,22]. Ryegrass grass and white clover are commonly combined in phytoremediation due to the synergy of using a dense rooting monocot with a nitrogen fixer [23]. Each pot contained 1 kg of soil. Pots were arrayed randomly in a 6 × 5 layout within a self-regulating greenhouse. Greenhouse conditions were set to 23 °C air temperature, 50% humidity, 16 h daylight/8 h night regime; 10 seeds of chicory and mustard were planted in each pot but were thinned to 3 plants per pot, and excess plants were removed upon germination. Fifty seeds were planted for the ryegrass grass varieties per pot; these treatments were thinned to twenty plants per pot at week 3. The number of seeds grown per pot was based on an estimation of the relative sizes of the plants in question. Mustard and chicory plants are much larger than ryegrass, which is why fewer individual seeds of the former were grown. The pot trial layout and plant species’ details can be found in the Appendix A.

### 2.3. Sampling for Microbiome Analysis

Soil samples were collected at weeks 0, 3 and 14 for bacterial and fungal microbiome analysis. Samples were taken from all treatment pots; however, only one monoculture, ryegrass treatment (RG1), was selected for sequencing, as there were no statistically significant differences in hydrocarbon degradation or biomass between ryegrass varieties. The treatments selected for microbiome analysis were as follows: control (C), mustard (M), ryegrass (RG1), ryegrass and white clover mix (RGWC) and chicory (Ch). Since a sample was taken from each pot, this resulted in 5 replicates per treatment, but multiple replicates were not taken from any single pot. The soil sampling procedure is outlined below.

At week 0, prior to pot allocation, the bulk soil was thoroughly homogenised to ensure uniformity before subsampling 5 replicates into 2 mL tubes. At weeks 3 and 14, soil samples were collected directly from each individual pot. To minimise contamination from the uncontaminated topsoil layer, a portion of this clean layer was carefully scraped away to expose the underlying contaminated soil. A composite sample, consisting of three subsamples, was taken from three different depths: top (1–3 cm), middle (5–7 cm) and bottom (10–12 cm). These subsamples were collected by digging a single vertical core to minimise the disturbance of the plant root system. Approximately 1 g of soil was extracted from each depth. The subsamples were then thoroughly mixed and transferred into 2 mL sampling tubes. All samples were immediately stored at −20 °C until the conclusion of the trial. Before DNA extraction, samples were removed from storage and allowed to thaw.

Some pre-treatment was performed prior to DNA extraction in order to maximise the recovery of the soil microbiome according to Martínez-Cuesta et al. [24]. A 2 g aliquot of soil was weighed and suspended in 2 mL of 0.85% NaCl solution in 15 mL tubes. The contents were vortexed at the maximum setting for 15 min then centrifuged at 300× *g* for 30 s. The supernatant was aliquoted to a 1.5 mL tube and pelleted at 8000× *g* for 10 min. This was repeated several times until all the supernatant was consumed. Thus, 60 µL/mL of lysozyme was added (1 mg/mL, prepared on the same day) and incubated for 1 h at 37 °C. Finally, the tubes were incubated at 80 °C for 5 min to inactivate the lysozyme then cooled to room temperature.

DNA extraction was completed using the FastDNA^®^ Spin Kit for Soil (MP Biomedicals Ltd., Santa Ana, CA, USA)) following the DNA kit procedure. Thus, 0.5 g aliquots of soil were used for the extractions, taken from each individual composite sample. Efficacy of each DNA extraction was determined by gel electrophoresis and analysis using a Qubit 4 fluorometer (Life Technologies Holdings Pte Ltd., Singapore). DNA extracts were sent to Novogene UK Limited (Cambridge, UK) for sequencing, targeting the bacterial 16S V3-V4 region using primers 5′-CCT ACG GGN GGCWGC AG-3′ and 5′-GAC TAC HVG GGT ATC TAA TCC-3′ (100 k tags per sample). For the fungal community, the sequencing targeted the 18S V4 region using primers 528F 5′-GCGGTAATTCCAGCTCCAA-3′, 706R 5′-AATCCRAGAATTTCACCTCT-3′ (50 k tags per sample). Bioinformatics was completed using the QIIME2 and Dada2 pipelines [25,26]. The 18S data were filtered to show sequences related to fungi only. Rstudio™ (RStudio, PBC, Boston, MA, USA; version 2023.12.1+402) was used to create data visualisation. Analyses were performed using R (version 4.3.2) with the following packages: DeSeq2 (version 1.40.2), phyloseq (version 1.44.0), QIIME2R (version 0.99.6) and ggplot2 (version 3.5.1) [27,28,29]. The SILVA 138 database, formatted for QIIME2 use (SILVA Release 138; https://www.arb-silva.de, accessed on 15 September 2024) was used for this analysis, which may contain some outdated bacterial taxa at the time of publication. A report on fungal microbiome results can be found in Appendix A.

### 2.4. Harvesting of Plants for Biomass

Dry biomass was determined for both roots and shoots at the conclusion of the trial. Shoots were harvested at the base of the plants. The samples were initially air dried in the greenhouse before transfer to an oven at 60 °C for a further 6 h. Roots were removed from the soil and cleaned prior to biomass analysis. The clean layer of soil was first removed from each pot and discarded. The remaining contaminated soil was gently removed from the roots by hand and collected for eventual PHC analysis. The residual soil was rinsed away from the roots in 0.9% NaCl solution. The roots were air dried in the greenhouse, and drying was completed in an oven at 60 °C for 6 h. Once at room temperature, the shoots and roots were weighed.

### 2.5. Determination of PHC Concentrations in Soil by GC-FID Analysis

Petroleum hydrocarbons were extracted from the soil using Soxhlet extraction. Once the roots had been removed from the contaminated soil for biomass analysis, the remaining soil was thoroughly mixed to ensure homogeneity. 50 g of this soil was aliquoted into individual weighing boats and dried overnight in an oven at 60 °C. After drying, samples were allowed to cool in a desiccator, and 10 g of each sample was placed into cellulose extraction thimbles with 10 g of sodium sulphate anhydrous to absorb moisture. Each sample was spiked with 100 mg/kg octafluoronaphthalene (OFN ≥ 99%, 248061—1 g; Sigma-Aldrich, St. Louis, MO, USA) surrogate spike. The purpose of the surrogate spike was to test the efficiency of the Soxhlet extraction. This was done by adding 1 mL of 1000 ppm OFN stock solution to each 10 g soil sample immediately prior to Soxhlet extraction. Then, 150 mL hexane and 150 mL acetone were used for the extractions, which took place in a waterbath set to 60 °C and lasted for 24 h. Once the extractions were completed, a rotavaporator was used to evaporate the hexane and acetone. The petroleum hydrocarbon residue was then re-dissolved in 2 mL analytical-grade hexane and filtered through a 0.45 µm syringe filter into 2 mL GC vials.

A PHC-certified reference material (CRM): Sigma-Aldrich TPH Mix (UST127 2000 µg/mL in dichloromethane:carbon disulphide (50:50)) was used both as a calibration standard and as a point of reference to test the efficacy of the GC method for eluting all the peaks clearly. The CRM consisted of 17 alkanes, ranging from carbon chain length of 8 to 40. The method used for this analysis achieved successful chromatography of all peaks with satisfactory separation and resolution. The GC method was as follows: Thermo™ Trace GC Ultra with an Agilent DB-PHC column; injector temperature: 280 °C; detector temperature: 300 °C; injection volume: 1 µL; Liner: splitless; carrier gas: nitrogen at constant pressure of 100 kPa; temperature program: 30 °C (hold for 3 min) ramp at 5 °C/min to 180 °C (hold for 4 min) ramp at 10 °C/min to 320 °C (hold for 5 min).

Calibration standards were made up using the PHC-certified reference material and were at concentrations of 10, 20, 40, 100, 200 and 400 mg/L. Octafluoronaphthalene standards were also made up and were at concentrations on 20, 50, 100 and 200 mg/L. Calibration curves and results were formulated using Microsoft Excel, version 2108 (Build 14332.21007); Microsoft Corporation, Redmond, WA, USA. It is very important to note that the soil tested contains a mixture of aromatics and aliphatics; however, only alkanes were used to formulate the calibration curves. Any aromatics extrapolated on these curves would have a stronger signal than equivalent alkanes (due to unsaturation and π electrons). As a result, the total petroleum hydrocarbon concentration determined by GCFID in this study is higher than that referenced in the soil conditions section due to a significant number of aromatics present in the soil. However, the primary goal is to assess the relative changes between treatments, so even though the absolute values might be inflated, the proportional difference should remain consistent and, therefore, meaningful.

### 2.6. Analysis of Heavy Metals by Atomic Absorption Spectroscopy

Samples of dried plant shoots, roots and soil were prepared for microwave digestion in nitric acid (69% concentration). This was done by weighing 0.5 g of dried plant or soil material into microwave digestion tubes followed by 7 mL concentrated nitric acid (69%) and 1 mL hydrogen peroxide (30%). The samples were placed in a microwave digestion system at 200 °C for 20 min with a ramp-up time of 10 min and a ramp-down time of 1 h at 40 °C/min. The digested solutions were filtered, made up to 50 mL in volumetric flasks and analysed using an atomic absorption spectrophotometer for copper, lead and zinc. The data were used to calculate bioconcentration factor (BCF) and translocation factor (TF). The formulae use to calculate BCF and TF, respectively, were:$$BCF=\frac{C_{Plant}}{C_{Soil}}\;TF=\frac{C_{Shoot}}{C_{Root}}$$

### 2.7. Statistical Methods

IBM^®^ SPSS^®^ Statistics 28 was used for all statistical analyses. Descriptive statistics confirmed that the data for total PHC and biomass followed a normal distribution. Therefore, one-way ANOVA tests were used to determine significant differences between treatments for total PHC concentration and biomass. Pearson’s correlation was used when comparing variables that followed a normal distribution. For variables that did not follow a normal distribution, including Shannon, Chao1, Simpson, and Pielou diversity indices, as well as fractional PHC data, Kruskal–Wallis tests were applied. Bonferroni corrections were used for all Kruskal–Wallis tests to adjust for multiple comparisons. Spearman’s rho and Kendall’s tau-b correlations were used to determine relationships between variables. Although some data were normally distributed, these non-parametric correlation methods were chosen to account for potential non-linear relationships when comparing normal and non-normal data.

### 2.8. Sequence Data Accession Numbers

The datasets presented in this study can be found in NCBI. The names of the repositories and accession numbers can be found at https://www.ncbi.nlm.nih.gov/ (accessed on 15 September 2024), PRJNA1054554 and PRJNA1055331.

## 3. Results

The primary aim of this work was to test the hydrocarbon degradation ability of several plant species and varieties in soils with mixed PHC and heavy metals. In addition, plant biomass, bacterial/fungal microbiomes and heavy metal bioconcentration and translocation factors were assessed.

### 3.1. Aerial and Root Biomass After 14 Weeks of Growth

Figure 1A,B show the aerial and root biomass generated after 14 weeks of growth in the contaminated soil. The dry aerial biomass of chicory was less than that of the RG1 and RG2 treatments by a statistically significant margin (*p* < 0.05) but not statistically different from mustard or RGWC. The aerial biomass of both mustard and RGWC was not statistically different from any other treatment. The large root biomass of chicory was composed mainly of a dense taproot, with limited contact with regions of the contaminated soil in the pot. Comparatively, the monocot ryegrass roots were spread evenly throughout the soil, offering more points of contact between the roots and the pollutants. Appendix A shows the average aerial, root and total biomass for each treatment, including standard deviations.

Mustard is also a dicot, like chicory; however, its primary taproot was smaller, and lateral roots were spread throughout the soil within the pot, offering more points of contact with contaminants and soil cohesion. These lateral roots were very narrow, fragile, and difficult to recover when compared to the other plants and may explain the low root biomass for mustard, which was lower than RG2 and chicory by a statistically significant margin (*p* < 0.05) but not different from RG1 or RGWC. There was no significant difference between RGWC, RG1, RG2 or chicory. An image showing the root structure of a ryegrass treatment (RG1) and mustard treatment (M) can be found in Appendix A. Total biomass was also determined, and statistical analysis was completed. Overall, mustard had the lowest mean total biomass, but this was only statistically significant when compared with the Abergreen ryegrass treatment (a 53.62% difference between the means of these two treatments). All of the biomass from the ryegrass and white clover mix was associated with ryegrass, as no clover germination was observed. This may be because the white clover was more sensitive to the metals in the contaminated soil. This meant that ryegrass in this treatment did not receive any increased nitrogen due to nitrogen fixation from white clover, as was the intention of this treatment. There was no statistically significant difference between any of the ryegrass treatments.

Biomass variables were correlated against all other variables to assess any possible positive or negative correlations. This was achieved using Kendall’s tau-b (τ_b) and Spearman’s rho (ρ). A significant positive correlation was observed between aerial biomass and root biomass for both correlation coefficients (τ_b = 0.654, *p* < 0.05; ρ = 0.837, *p* < 0.05), suggesting a strong relationship between above-ground and below-ground biomass production in phytoremediation systems. Spearman’s rho produced higher correlation values than Kendall’s tau-b across all variables, as it is generally more sensitive to larger sample sizes and rank differences. However, both methods indicate the same direction and significance of relationships, confirming the robustness of these correlations. The same pattern was observed for aerial biomass and total biomass (τ_b = 0.837, *p* < 0.05; ρ = 0.867, *p* < 0.05) and for root biomass and total biomass (τ_b = 0.565, *p* = 0.028; ρ = 0.565, *p* = 0.028). Spearman’s rho analysis also showed a positive correlation between observed bacterial ASVs and both root (ρ = 0.524, *p* < 0.05) and aerial biomass (ρ = 0.524, *p* < 0.05), suggesting a potential relationship between bacterial community diversity and biomass production [30].

### 3.2. Reduction in Petroleum Hydrocarbon (PHC) Concentrations Under Microcosm Conditions

The PHC content of the soils after 14 weeks of rhizoremediation treatment and in the control soil was measured by GC-FID (Figure 2). The surrogate spike octafluoronapthalene (OFN) was first determined to assess PHC extraction efficiency. The theoretical value for the recovered OFN was 50 mg/L. The averaged recovered OFN was 60 mg/L OFN. Percent recovery was 120%, with a standard deviation of 29%. In terms of PHC degradation, mustard outperformed the control and chicory by a statistically significant margin (*p* < 0.05). There was no statistically significant difference between the control and any other treatments. However, there was a large amount of variability in PHC degradation within the replicates of the control treatment (between 3 and 81% degradation versus the initial concentration), making it difficult to interpret results when comparing to the plant treatments, which had less variation in final concentrations. At 14 weeks, the control (non-planted soil) had an average degradation of 40% versus the initial level, whereas the Abergain (RG1) and Abergreen (RG2) ryegrasses improved on this by a further 16%. A slight inhibition in PHC degradation was observed with the chicory. Mustard performed better, with 28% and 12% increases on the control and the ryegrasses, respectively. Using Pearson’s correlation, no link was found between plant biomass and degradation of the PHCs, either within or between the treatments. Alternative explanations for the contributing factors to hydrocarbon degradation are explored in the Discussion section.

Table 1 shows the average percentage degradation of hydrocarbons based on carbon chain length. It can be seen that as carbon chain length increases, degradation success decreases. However, mustard maintains the highest degradation for every faction. Table 1 shows the fractional averages and standard deviations of the high-molecular-weight compounds c08–c40. There was no statistically significant difference between the mustard and ryegrass treatments. From Table 1, it is clear from c20 to c40 that the control is not statistically different from any other treatments, even though the other treatments are statistically different from each other occasionally. This is because the control had a very high deviation between replicates.

Figure 3 shows some of the results obtained from analysis of the hydrocarbons. Most of the hydrocarbons are between c20 and c40 (Table 1). Hydrocarbons in the c20–c24 range are reduced in all treatments, including the control by a statistically significant amount, leaving most of the remaining hydrocarbons in the c26–c40 range. Except for the c8–c10 range, mustard had the lowest mean concentration for every fraction of hydrocarbons. Table 1 also illustrates the significant differences for each fraction for every treatment. Of the total 16 fractions that were analysed, mustard reduced 13 out of 16 fractions by a statistically significant margin from the initial level (*p* < 0.05). Mustard was significantly lower from the control in 3 out of the 16 fractions, chicory in 8 out of 16 fractions, RG1 (Abergain) in 2 out of 16, rRG2 (Abergreen) and RGWC in 12 out of 16 fractions (*p* < 0.05). Mustard was the only treatment to outperform the control in several hydrocarbon fractions but was not significantly different from other treatments.

### 3.3. Heavy Metal Concentrations in Soil, Roots, and Shoots

Table 2 shows the concentration of heavy metals detected in the soil and plant roots and plant shoots. The bioconcentration factors and translocation factors were calculated and averaged. Bioconcentration factors greater than one indicate that there is a higher concentration of metal in the plant than in the soil. Translocation factors greater than one indicate that there is a higher concentration of metal in the aerial parts of the plant than the roots.

In the case of lead, no plant treatment achieved a BCF or TF greater than one. Mustard had the lowest BCF for lead, which was statistically significant when compared with ryegrass 1 and rye grass + white clover (*p* < 0.05). The BCF between RG1 and RG2 was also found to be statistically significantly different (*p* < 0.05) There was no statistically significant difference between the TF for any treatment pertaining to lead. For copper, chicory had the lowest mean BCF, which was statistically significant compared to mustard and ryegrass 2 (*p* < 0.05). There was also no significant difference between any treatment relating to TF for copper. Finally, for the BCF of zinc, there was no statistically significant difference between treatments, nor did any treatment achieve a BCF greater than 1. Mustard had the highest TF for zinc by a statistically significant margin compared to RG2 and RGWC (*p* < 0.05), and there were no other treatments that had significant differences for BCF or TF (*p* < 0.05).

### 3.4. Bacterial Community Dynamics and Diversity During Phytoremediation

The bacterial and fungal communities present in the soil were determined prior to the trial, mid trial and at the end of the trial. Amplicon sequence variants (ASVs) were used rather than operational taxonomical units (OTUs) as they provide greater resolution and better reproducibility. The bacterial 16S amplicons were sequenced at a sampling depth of 100,000 raw tags. To achieve consistent sequencing depth for all samples, the data were rarefied to a depth of 95% of the minimum sample depth in this dataset, which resulted in just under 87,000 raw tags. At this sampling depth, the rarefaction curves plateau before the maximum sampling depth, indicating the majority of species present were captured at this sampling depth (Appendix A and fungi Appendix A). Information on observed ASVs, Chao1, Shannon, Simpson’s inverse, weighted/unweighted unifrac and Bray–Curtis distances can be found in Appendix A. The mustard-containing soil had the highest mean observed ASVs at week 14. This was significantly higher than RG1 (*p* < 0.05). The statistical significance of the observed ASVs, Chao1, Simpson’s inverse and Shannon index can be found in Appendix A.

Figure 4 shows the major bacterial genera detected in each treatment and the pre-trial soil at the three sampling times. Organisms with a relative abundance of less than 0.5% were considered below the detection threshold and were classified as ‘Other’. For the pre-trial soil, the most abundant bacterial phyla in the soil were Proteobacteria (59%), Actinobacteriota (26%) and Bacteroidota (4%). At week 3, the abundance of Proteobacteria in all treatments was detected to be between 79 and 88%, Bacteroidota was 4 and 9% and Actinobacteriota abundances were 1.6 and 3.8%. By week 14, Proteobacteria abundance was detected within a range of 71–80% and Actinobacteriota was 4.8–10%. Several other phyla such as Patescibacteria (3–6.5%), Firmicutes (1–2.3%) and Acidobacteriota (0.7–1.8%) rose above the detection threshold for the first time in all treatments.

At the family level, *Caulobacteraceae* (14.1%), *Xanthomonadaceae* (12.4%), *Rhizobiaceae* (10.1%), *Microbacteriaceae* (8.7%), *Xanthobacteraceae* (7.4%) and *Pseudomonadaceae (6.8%)* were detected in the pre-trial soil. At week 3, the average range of relative abundance was *Pseudomonadaceae* (18.6–34.3%), *Xanthomonadaceae* (22.2–31.3%), *Caulobacteraceae* (4.1–10.6%), *Rhizobiaceae* (5.1–7.7%), *Xanthobacteraceae* (1.3–2.3%) and *Microbacteriaceae* (0.9–1.6%). In addition, *Alcaligenaceae* (4.3–6.8%), *Acidithiobacillaceae* (1.9–3.2%) and *Flavobacteriaceae* (0.6–2.9%) were above the 0.5% detection threshold. At week 14, the relative abundance of these families was detected to be *Xanthomonadaceae* (20.5–25.2%), *Pseudomonadaceae* (7.1–15.8%), *Acidithiobacillaceae* (6.9–10.1%), *Microbacteriaceae* (2.5–4.5%), *Rhizobiaceae* (3.1–4.0%), *Caulobacteraceae* (2.5–4.0%), *Alcaligenaceae* (2.8–3.6%)*, Xanthobacteraceae* (2.3–2.4%), and *Flavobacteriaceae* (0.7–1.5%).

Finally, at the genus level for week 0 (pre-trial) soil, *Pseudomonas* (6.2%), *Olivibacter* (2.3%) and *Pseudoxanthomonas* (0.5%) were the only genera conclusively identified at the genus-level resolution. Unclassified genera above the 0.5% abundance threshold accounted for 43% and other genera below the threshold accounted for 48.1%. At week 3, several new genera rose above the detection threshold; these were *Stenotrophomonas* (11.7–16.9%), *Achromobacter* (3.2–6.3%), KCM*-B-112* (1.9–3.2%), *Flavobacterium* (0.6–2.8%), *C1-B045* (0.6–1.3%), *Luteimonas* (0.3–1.4%) and unclassified *Saccharimonadales* sp. (0.4–0.7%). By week 14, the detected genera were *KCM-B-112* (6.9–10%), *Stenotrophomonas* (2.9–7.4%), *C1-B045* (2.4–5.6%), unclassified *Saccharimonadales* sp. (1.4–3.3%), *Achromobacter* (2.4–2.8%) and *Olivibacter* (0.3–0.6%).

Figure 5A,B show results for the bacterial Shannon and Chao1 index, respectively. There was no statistically significant change in the Shannon index from the pretrial level compared to any of the treatment soils at week 3. However, by week 14, all treatments saw statistically significant increases in the Shannon index compared to the pre-trial. There was no statistically significant difference between the control or any of the treatments at the week 14 sampling time (*p* < 0.05). The control and Abergain treatments each had outliers for the Shannon index (as seen in Figure 5; these were treatment replicates in which there was very little or very large percentage PHC degradation). When looking at the Chao1 index, however, there was a statistically significant increase in alpha diversity by week 3 for all treatments compared to the pre-trial soil, but there was no significant difference between the treatments themselves (*p* < 0.05). There was no statistically significant increase in the Chao1 index between weeks 3 and 14 for any treatments, nor was there a significant difference between any treatment at week 14. Chao1 utilises the taxa in a sample with fewer counts and uses them to estimate the number of taxa that are truly in the sample but are not observed; this may explain the more prominent significant differences between treatments compared using Chao1 compared to the Shannon index. The Simpson’s inverse was also determined, and, like the Shannon and Chao1, there was no significant difference between any of the treatments by week 14 (Appendix A).

Figure 6 is a principal coordinate analysis that shows the bacterial beta diversity for each treatment. Figure 6 uses the Bray–Curtis distances to visualise differences in microbial communities. This metric does not take phylogenetics into account. In this case, the clusters are separated based on both treatment and sampling time. Most of the replicates for treatments were very similar. There was a large shift in bacterial communities between week 3 and 14. Additionally, by week 3, most of the treatments were clustered together with the exception of two mustard replicates; however, by week 14, there was a clear separation between the control and plant treatments. This may demonstrate the impact of the rhizosphere effect on the microbial communities. Appendix A utilise Weighted and Unweighted UniFrac respectively (fungal Weighted and Unweighted UniFrac can be found in Appendix A). This analysis considers both branch length on the phylogenetic tree and bacterial abundance when illustrating diversity similarity. When those data are looked at from this point of view, there is very little difference between or within any of the treatments. The combination of Bray–Curtis and weighted unifrac metrics suggests that though the bacteria found in each treatment are different species, they are phylogenetically closely related. However, this does not necessarily indicate that they belong to the same genus; rather, they may be from different genera within the same family or a higher taxonomic group.

Figure 7A shows the results for differential abundance analysis of the pre-trial versus the control treatment at week 14 and the control at week 14 verses mustard, chicory and ryegrass treatments at week 14 (Figure 7B, Figure 7C and Figure 7D, respectively). Figure 7A shows the major genera that became prominent in the control treatment, with several genera associated with hydrocarbon-contaminated soils, such as *KCM-B-112*, *C1-B045*, *Stenotrophomonas*. unclassified *Saccharimonadales* sp. (7–8 log2 fold change) and *Pseudomonas* (>10 log2 fold change) being enriched. Many genera were depleted in the control, such as *Xanthomonas*, *Streptomyces*, *Flavobacterium*, and *Oliviobacter* (all >5 log2 fold change). The mustard treatment (Figure 7B) saw significant enrichments at the genus level in abundances mainly in *C1-B045*, *Hydrogenophaga* (1–2 log2 fold change), unclassified *Saccharimonadales* sp. and *LWQ8* (4–5 log2 fold change).

The largest enrichments were in *Pedobacter* and *mle1-27* (>5 log2 fold change). In order of largest to smallest log2 fold change, *Comamonas*, *Petrimonas*, *Geobacter*, *Lysobacter* and *Azospirillium* were depleted (between −4 and −10 log2 fold change) in mustard compared to the control at week 14. For the Abergain ryegrass grass (Figure 7C), the main enriched genera were *Nordella*, *Iamia*, *Dyadobacter*, *Hydrogenophaga*, *Niastella, Terrimonas*, *Chiayiivirga* (1–2 log2 fold change) and *LWQ8* (log2 fold change of 4). *Neisseria,* which had the largest log2 fold change between ryegrass and the control, is a genus of bacteria known for colonising the mucosal surfaces of animals and likely a contaminant. *Comamonas*, *mle1-27*, *Geobacter*, *Petrimonas* and *Azospillium* were all depleted in ryegrass compared with the control. Finally, chicory (Figure 7D) saw enrichment in *Iamia*, *Hydrogenophaga*, *Terrimonas, Niastella* and *C1-B045* genera; however, the changes in differential abundance were much smaller when compared with mustard and ryegrass. *Parvibaculum*, *Lysobacter*, *Pedobacter* and *Stenotrophomonas* were some of the main genera depleted in chicory compared with the control. The as-yet uncultured clade, *mle1-27*, and genus *Pedobacter* were depleted in ryegrass and chicory, respectively, when compared to the control. However, both of these were enriched when comparing mustard with the control.

## 4. Discussion

This study demonstrated the rhizoremediation potential of several plant species through a short-term microcosm-scale experiment. The control treatment (biostimulation) led to a 40% reduction in petroleum hydrocarbon (PHC) levels, with degradation ranging from 3% to 81% after 14 weeks. This figure is notably higher than those typically reported in the literature, where hydrocarbon degradation rates usually range between 20 and 30% [16,18,31]. The enhanced degradation may be attributed to the enrichment of hydrocarbon-degrading genera such as *KCM-B112* and *C1-B045*, which exhibited a 7–8 log2 fold increase compared to pre-trial levels. Several additional factors may have contributed to the variation in PHC degradation observed in the control treatments, including antagonistic microbial interactions, localised fluctuations in oxygen and water availability due to soil compaction, and microbial lag phases or dormancy [23,32,33].

The considerable variability within the control undoubtedly influenced the statistical significance of plant treatments when compared to the control, highlighting an important consideration when assessing the efficacy of the plant-based remediation. The lack of statistically significant differences in hydrocarbon degradation between ryegrass, chicory, and the control is likely due to unexpectedly high degradation in some control replicates. The randomised arrangement of the pot trial was carefully examined to determine whether pot placement influenced the variability observed in the control results. However, no discernible pattern emerged to suggest that certain locations within the layout experienced superior degradation. The potential heterogeneity of PHC in the soil was also considered, but since the soil for all treatment pots was mixed in bulk and allocated after homogenisation, any lack of uniformity would have affected all treatments equally. Additionally, despite similar alpha and beta diversity among control replicates, subtle but important microbial community differences may not have been captured.

Although the introduction of plants can disrupt microbial equilibrium with pollutants in pore water, this is unlikely to have been a factor in this study. The pre-treatment process, which involved air drying and sieving the soil through a 2 mm mesh, would have eliminated pre-existing microbial heterogeneity. Consequently, upon rewetting at the start of the experiment, microbial populations in all treatments, including the control, would have been undergoing adaptation and re-establishment rather than maintaining a prior equilibrium. Lastly, inconsistencies in degradation rates may have arisen if the slow-release fertiliser was not uniformly homogenised when applied to the pots, leading to variable microbial responses [34].

There was no correlation between plant biomass and hydrocarbon degradation in this study. Compared to findings in the literature, the hydrocarbon degradation rate for ryegrass in this study was slightly higher than those reported in similar trials. However, it was consistent with degradation rates observed when additional organisms, such as *Bacillus subtilis* and *Acinetobacter radioresistens*, were introduced to enhance phytoremediation [35]. This may suggest that microbes already present in the contaminated soil contributed to a similar effect to that observed by Tang et al. [35]. Root exudate dynamics, including the volume released, exudate composition, and spatial distribution within the pots, are key factors influencing degradation processes but were not examined in this study [9].

Although direct degradation via root exudates may have played a role in hydrocarbon breakdown, other mechanisms, including co-metabolism, biosurfactants, and rhizosphere effects, are likely to have been more significant contributors to the degradation and bioavailability of certain high-molecular-weight hydrocarbons [9]. The phytoremediation treatments, particularly mustard, exhibited a higher average degradation of high-molecular-weight hydrocarbons. Differential abundance analysis was conducted to assess variations in bacterial genera between the plant treatments and the control. Analysis of the soil microbiomes identified several bacterial genera that may have contributed positively to hydrocarbon degradation. Among these, *KCM-B-112* and *C1-B045* were the most notable, as they were detected in all treatments, including the control. *KCM-B-112* has been widely reported as a dominant genus in petroleum hydrocarbon-contaminated soils [36,37,38]. *C1-B045* has been found to degrade methylcyclohexane in deep-sea-surface sediments. One strain from the Indian Ocean had its genome sequenced and was found to possess two copies of alkane monooxygenase-like genes, as well as genes encoding cyclohexanone monooxygenase and one gene encoding 6-hexanolactone hydrolase [37,38]. Several recent studies have also linked *C1-B045* to the microbial degradation of light crude oil [39], polycyclic aromatic hydrocarbons [40], and short-chain *n*-alkylcyclohexanes [40,41].

Among the plant treatments, mustard achieved the highest average hydrocarbon degradation, with a 68% degradation rate. It was also the only treatment to significantly outperform the control. While it is not possible to precisely quantify the time required for complete soil remediation, since only high-molecular-weight hydrocarbons remained, full remediation is estimated to be achievable within 20–40 weeks, given that the mustard plants had not yet reached full growth and could have persisted for several months longer. Compared to the control, mustard exhibited differential increases in several genera associated with hydrocarbon degradation. These included *C1-B045, Hydrogenophaga, Saccharimondales* sp., and *Pedobacter*. *Hydrogenophaga* strains have been linked to the degradation of hydrocarbons, pyrene, and benzo[a]pyrene under both anaerobic and aerobic conditions [37,38,42,43,44]. They also play a role in reducing hexavalent chromium to Cr(III) and possess plant growth-promoting properties [42,43]. Species in the *Saccharimondales* order can tolerate heavy metals, degrade polycyclic aromatic hydrocarbons (PAHs), and exhibit synergistic effects with nitrogen cycling genes [45,46]. *Pedobacter* is a key member of alkane-degrading bacterial communities [44]. The most enriched bacterial taxon in mustard was the uncultured clade *mle1-27*, which showed a log2 fold enrichment of 9–10. It is considered a putative bacterial predator. Although it lacks a strict prey preference, it has been strongly correlated with pollutant removal rates in activated sludge, as well as with polyphosphate-accumulating and -denitrifying bacteria. As such, *mle1-27* may play a trophic role by supporting microbial communities involved in hydrocarbon degradation rather than directly preying on hydrocarbon-degrading bacteria [47]. Mustard plants may serve as a useful alternative to ryegrass, as their deep-rooting system, rather than a dense, shallow root system, may be advantageous in situations where contaminants are located deeper than ryegrass roots can reach. Additionally, mustard can immobilise nitrogen within its tissues, which can later be recycled into the ecosystem upon decomposition at the end of its growth period [48]. However, unlike the perennials, ryegrass and chicory, mustard is an annual plant. This means that if remediation is not completed within its lifecycle, reseeding the following year would be necessary, significantly increasing the financial costs and time required for the remediation process.

The ryegrass treatments also exhibited greater average degradation than the control (16%, range: −5–47%); however, this difference was not statistically significant. This result aligns with findings from longer trials lasting over 20 weeks [16,35]. In terms of differential abundance of bacteria for ryegrass, the most notable increases in bacterial abundance were observed in *Hydrogenophaga, Nordella*, *Dyadobacter*, and *Terrimonas*. *Nordella* has been linked to PAH degradation in the rhizosphere of *Leymus chinensis* (Chinese ryegrass) [49]. *Dyadobacter* is a genus containing multiple species known for hydrocarbon degradation [50], while *Terrimonas* has been associated with benzo[a]pyrene degradation in soil [51]. The significant enrichments in bacterial abundance between chicory and the control were observed for *Hydrogenophaga*, *Terrimonas*, and *C1-B045*; however, these changes were not as pronounced as those in mustard or ryegrass.

Chicory was the poorest-performing phytoremediation treatment, with an average degradation rate 17% lower than that of the control (range: 11–27%). Its poor performance may be attributed to several factors. Firstly, its root system had fewer contact points with the bulk soil. Despite having a relatively large root biomass, chicory’s roots were primarily concentrated in a dense taproot with very few lateral roots, limiting the extent of soil benefiting from rhizoremediation [22]. However, these factors alone do not fully explain why chicory underperformed compared to the other plant treatments. To obtain a better understanding of chicory’s poor performance, we investigated the microbial community. Compared to the control, several bacterial genera associated with enrichment in hydrocarbon-contaminated soils and/or hydrocarbon degradation, including *Pedobacter, Lysobacter*, *Rhodanobacter*, and *Massilia*, were found to be depleted in the chicory treatment. Conversely, the genera enriched in chicory—*Lamia*, *Hydrogenophaga*, *Terrimonas*, *Niastella*, and *C1-B045*—are also linked to hydrocarbon-contaminated soils, but their enrichment was marginal (1 to <4 log2 fold change) [24,36,37,38,44,50,52,53]. Certain root exudates can exert selective pressures on the soil microbiome, potentially disadvantaging microbes involved in hydrocarbon degradation. If this was the case for chicory, it may explain its inferior hydrocarbon degradation performance. Such interactions can vary depending on plant species and environmental conditions. However, as environmental conditions were controlled in this experiment, it is likely that the characteristics of chicory itself influenced the observed outcomes [54].

When comparing differential abundance between plant treatments, two genera that were enriched in mustard, *mle1-27* and *Pedobacter*, were depleted in ryegrass and chicory, respectively. *Mle1-27* is a bacterial predator, while *Pedobacter* possesses hydrocarbon degradation capabilities. Other bacterial genera that were depleted across all plant treatments, such as *Flavobacterium*, typically thrive in wet environments [47]. As such, competition for water with plant roots may have reduced their abundance. As demonstrated above, several bacterial genera that were significantly more abundant in mustard compared to the control have strong links to hydrocarbon degradation, suggesting they may play a key role in the degradation success observed in mustard treatments.

Several bacterial genera were depleted in the plant treatments compared to the control, with anaerobic bacteria being most affected. One possible explanation is that plant roots altered soil aeration dynamics. While living roots absorb oxygen from the soil, they also indirectly enhance aeration by reducing soil water content, increasing air-filled pore spaces. This shift favours aerobic bacteria, allowing them to outcompete anaerobic bacteria [11]. In contrast, control soils retained moisture for longer, creating more anaerobic microenvironments where anaerobic bacteria could thrive [15]. Some of the anaerobic genera identified included *Proteiniphilum*, *Geobacter*, and *Petrimonas*. The primary benefit of phytoremediation over biostimulation was observed in the degradation of high-molecular-weight PHC fractions, particularly in the mustard treatment, which exhibited a 60% reduction in hydrocarbons within the C34–C36 range by week 14 (Table 1). This suggests that mustard may be a strong candidate for the rhizoremediation of recalcitrant or weathered hydrocarbons. In contrast, chicory was the least effective in degrading high-molecular-weight hydrocarbons. Additionally, *Stenotrophomonas* and *Achromobacter* also showed an increase in abundance for all treatments, particularly at week 3, decreasing again by week 14. This decrease was more pronounced in the plant treatments. This can be seen in Figure 4. Species in these genera have been associated with hydrocarbon-contaminated environments. *Stenotrophomonas* species, such as *S. maltophilia*, have been linked to hydrocarbon degradation via biosurfactant production, which enhances contaminant bioavailability [55,56]. Similarly, *Achromobacter* species, such as *A. xylosoxidans*, have been detected in petroleum-contaminated soils, though their direct role in hydrocarbon degradation remains unclear. Their enrichment in this study may be a response to rhizosphere conditions or an opportunistic adaptation to the contaminated environment [57,58]. Additionally, species in both *Achromobacter* and *Stenotrophomonas* both contain metal resistance genes, with the ability to tolerate metals, such as cadmium, copper, zinc and arsenic. Both genera can support phytoremediation by reducing metal toxicity in the rhizosphere, improving plant survival in contaminated soils and potentially enhancing metal uptake into plant tissues. This tolerance to heavy metals may also be a factor in the increased abundance of these genera at week 3. The versatility of *Achromobacter* and *Stenotrophomonas* in hydrocarbon and metal remediation suggests they may be good bioaugmentation candidates [59,60].

Fungal community analysis was also conducted and is presented in Appendix A. No major conclusions could be drawn from the differential abundance analysis of fungal communities. Chicory exhibited no significant changes compared to the control, while mustard showed an increase in two *Ascomycota* genera and a depletion in *Arthrobotrys*. Ryegrass exhibited a larger increase, with five *Ascomycota* genera showing enrichment, likely due to the greater substrate availability resulting from its larger root biomass. While both ryegrass and chicory are capable of forming arbuscular mycorrhizal associations, none were observed in the microbiome data. Mustard, being a member of *Brassicaceae*, does not form mycorrhizal relationships [61].

A significant increase in bacterial alpha diversity was observed across all treatments, a trend that has been linked to a reduction in petroleum hydrocarbons in soil through plant-assisted bioremediation [24]. In contrast, fungal alpha diversity did not increase, which may be attributed to the proliferation of species from the family *Herpotrichiellaceae*, which became dominant in all treatments. This family includes species with hydrocarbon degradation potential [62]. However, given that it comprises approximately 270 species, including potential human pathogens and plant endophytes, it remains unclear whether this organism directly contributed to the remediation process.

Like other members of *Brassicaceae*, mustard has demonstrated phytoextraction capabilities in heavy metal-contaminated soils [17,21]. The metal analysis suggests that mustard may be effective in extracting copper but much less effective in extracting lead or zinc, which may be due to toxicity or limited bioavailability. However, none of the plant treatments showed any visible signs of phytotoxicity due to the presence of heavy metals, though their presence may have been a factor in preventing white clover from germinating. These results suggest mustard was less effective in accumulating zinc when compared with other studies on mustard, which found cadmium, copper and zinc accumulation occurred in mustard plant tissues without evidence of phytotoxicity. Additionally, mustard has been reported in the literature to extract other metals such as arsenic and platinum. This tolerance to heavy metals may have played a role in the enhanced petroleum hydrocarbon bioremediation observed in this experiment. We were not able to determine why mustard’s phytoextraction capacity was less than what was observed in previous studies, but seeing that phytoextraction with mustard can be improved by specific microorganisms may imply that the microbiome in this soil was not appropriate for efficient phytoextraction. Further investigation is needed to fully understand the phytoextraction potential of mustard, particularly since its efficacy will depend on the specific metal complexes present in different soil environments [17,21].

## 5. Conclusions

It can be concluded that on a microcosm scale, seeding with mustard is a promising approach for rhizodegradation, as a significant decrease in petroleum hydrocarbons was observed when compared with the control. Analysis of the soil microbiome provides evidence that this may be linked to the presence of bacterial genera associated with hydrocarbon degradation, possibly promoted by the rhizosphere effect in addition to the presence of *mle1-27* clade and *KCM-B-112*, which show strong correlation with pollutant degradation in the literature. Increased abundances in these key genera may explain why mustard was the only treatment to outperform the control by a statistically significant margin. Ryegrass has a history of use in phytoremediation and provides benefits such as helping microorganisms migrate through the soil and the formation of dense root structures, providing substrates for rhizospheric microorganisms, boosting their abundance. Despite this, none of the ryegrass treatments used in this trial outperformed the control by a statistically significant margin, though this may change over a longer treatment time. It is also important to note that each of these plant species may perform differently in alternative soil conditions (pH, organic matter content, soil texture etc.), meaning mustard is not necessarily the best option in all cases. Ultimately, our hypothesis was supported as significant differences were found for hydrocarbon degradation between treatments. This is highlighted by the mustard treatment, which has a statistically better degradation percentage than both the control and chicory treatments. Differences in microbial communities were also observed, especially relating to alpha diversity and enrichment/depletion of some organisms, but most of these changes were only significant by week 14. However, using the broader lens, beta diversity metrics of bacteria showed little difference between treatments but large differences between sampling times. Differences in beta diversity were more pronounced for fungi. Overall, our findings underscore the importance of plant selection in phytoremediation strategies and highlight the potential of mustard as a valuable addition to remediation efforts. Further research is warranted to explore the full extent of mustard’s phytoextraction capabilities and its broader applicability in contaminated environments.

## Figures and Tables

**Figure 1 microorganisms-13-00848-f001:**
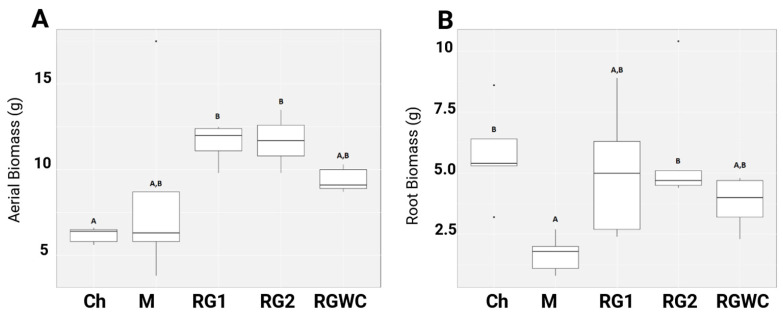
Dry biomass of (**A**) aerial and (**B**) root biomass from phytoremediation trial 2. Treatment vs. weight (g). Replicates of each plant treatment *n* = 5, standard deviation used for error bars. M: mustard, RG1: Abergain (ryegrass), RG2: Abergreen (ryegrass), RGWC: ryegrass grass and white clover, Ch: chicory. Letters above boxplots denote statistical significance, treatments that share the same letter are no significantly different (significant threshold: 0.05).

**Figure 2 microorganisms-13-00848-f002:**
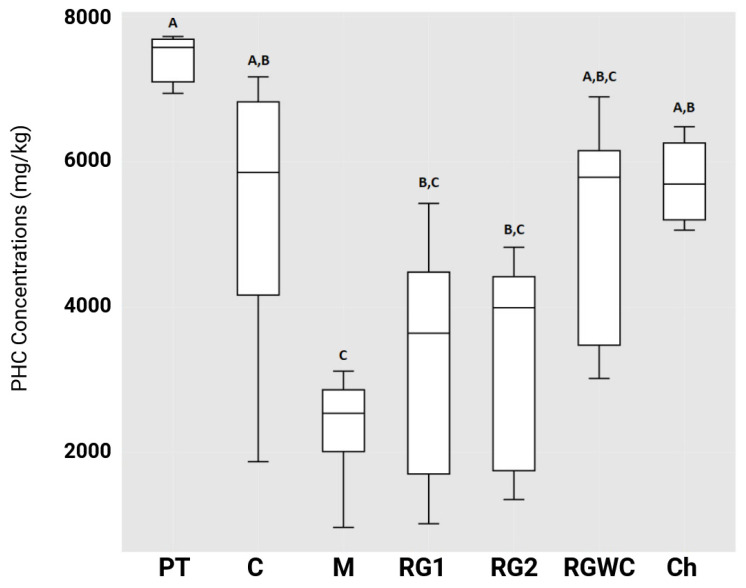
Petroleum hydrocarbon (PHC) levels remaining in the soil after 14 weeks. Treatments that share the same letter are not significantly different (replicates *n* = 5). PT: Pre-trial concentration, C: control, M: mustard, RG1: Abergain (ryegrass), RG2: Abergreen (ryegrass), RGWC: ryegrass and white clover, Ch: chicory.

**Figure 3 microorganisms-13-00848-f003:**
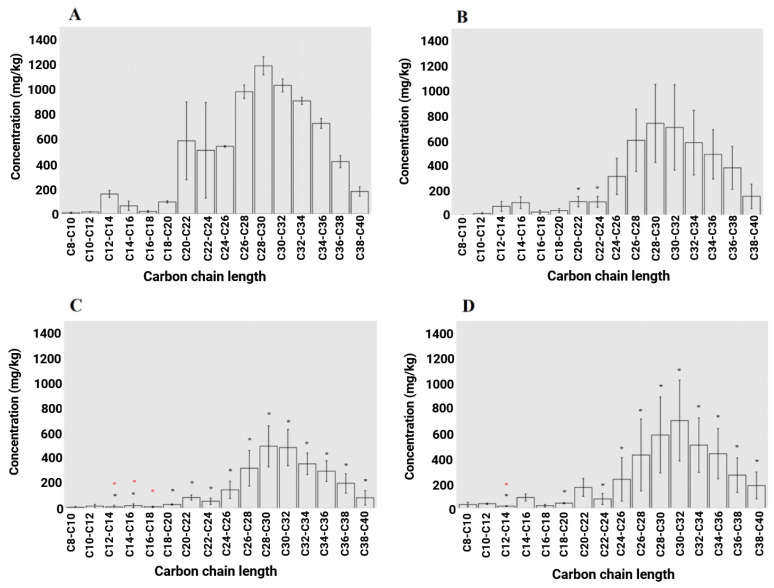
Average level of hydrocarbons fractions detected in soil for (**A**) the initial level at the start of the experiment, (**B**) control after 14 weeks, (**C**) mustard after 14 weeks and (**D**) ryegrass Abergain after 14 weeks. Results are broken into carbon chain length. N = 5, error bars are the standard deviation between replicates. A black asterisk (*) indicates a statistically significant decrease in hydrocarbon concentrations between the treatment and pre-trial levels. A red asterisk (*) denotes a statistically significant decrease compared to the control treatment (*p* < 0.05).

**Figure 4 microorganisms-13-00848-f004:**
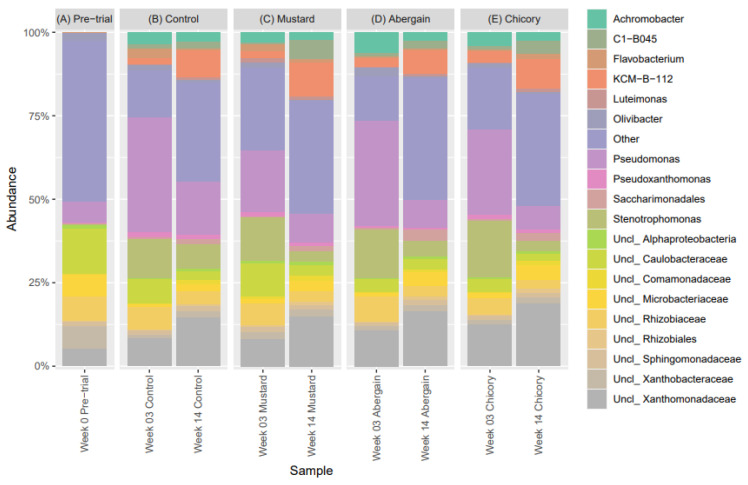
Major genera of bacteria found at each of the three sampling times: week 0, week 3 and week 14. Samples are grouped by treatment (top) and sampling date (bottom). (*n* = 5).

**Figure 5 microorganisms-13-00848-f005:**
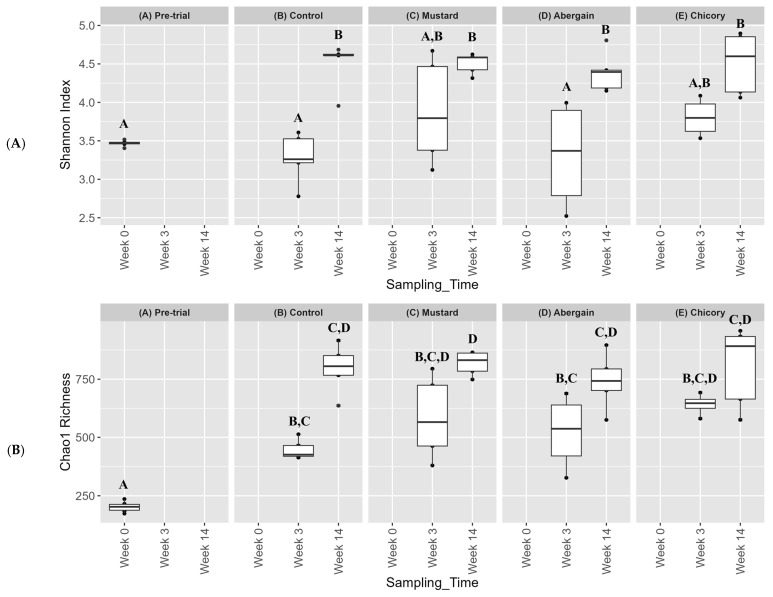
Bacterial alpha diversity using the Shannon (**A**) and Chao1 (**B**) index values. Samples are grouped by treatment and all three sampling times are represented. Samples that do not share the same letter are significantly different (*p* < 0.05) Shannon index (normal) and Chao1 (non-normal) were determined using the Tukey test and Kruskal–Wallis test, respectively (*n* = 5).

**Figure 6 microorganisms-13-00848-f006:**
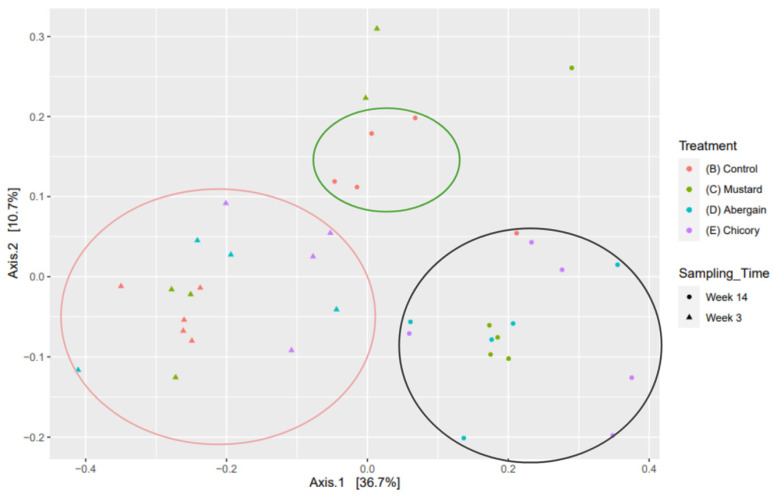
Principal coordinate analysis using Bray–Curtis distances for bacteria, a qualitative beta diversity analysis. All replicates included (*n* = 5). Pre-trial data points were removed to allow for greater resolution when viewing the treatments.

**Figure 7 microorganisms-13-00848-f007:**
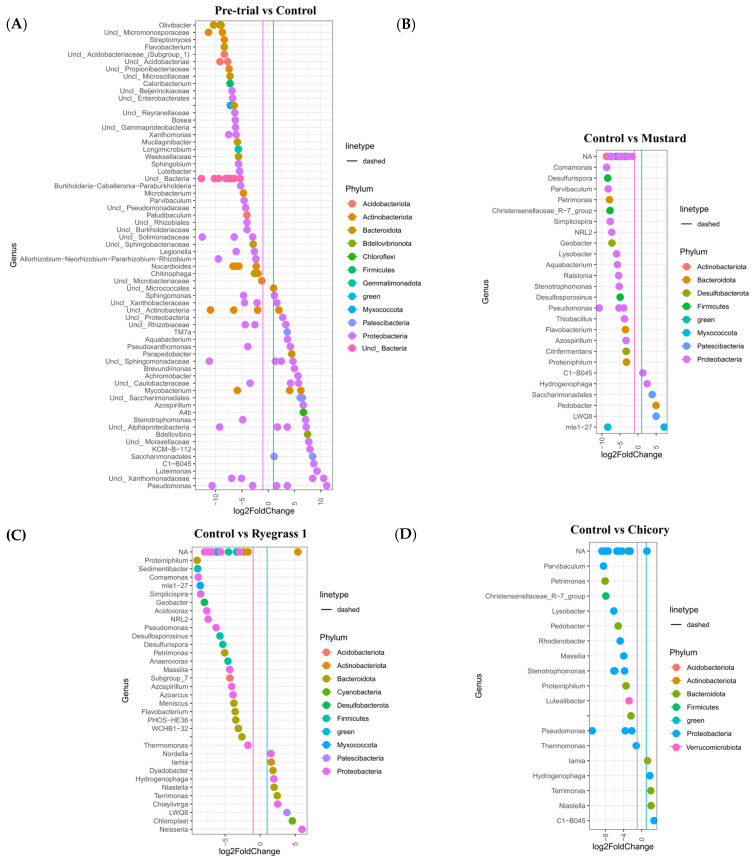
Differential abundance analysis of bacterial ASVs. The graphs show the significant differences at genus level between the pre-trial soil and control (**A**) and the control and different treatments (**B**–**D**) at the week 14 sampling time (*p* adjusted value < 0.01). Colours denote phylum and the x axis represents the log2 fold change.

**Table 1 microorganisms-13-00848-t001:** The average percent degradation of the hydrocarbon fractions for each treatment (*n* = 5) after 14 weeks.

C-Chain Length	Control	Mustard	Ryegrass (Abergain)	Ryegrass (Abergreen)	Ryegrass + White Clover	Chicory
c8–c10	64 ± 85	30 ± 102	0 ± 56	0 ± 90	0 ± 46 *	0 ± 54 *
c10–c12	32 ± 81	56 ± 113	39 ± 54	28 ± 56	0 ± 49	0 ± 40
c12–c14	57 ± 54	95 ± 158 *	94 ± 74 *	83 ± 88	85 ± 114	87 ± 56
c14–c16	0 ± 47	75 ± 74 *	17 ± 39	24 ± 39	18 ± 33	44 ± 25
c16–c18	0 ± 55	52 ± 52 *	19 ± 45	15 ± 57	0 ± 44	0 ± 32
c18–c20	58 ± 40	72 ± 17	63 ± 12	57 ± 23	45 ± 27 *	44 ± 34
c20–c22	81 ± 37	86 ± 23	78 ± 24	72 ± 45	68 ± 32 *	68 ± 17 *
c22–c24	85 ± 35	91 ± 45	89 ± 72	87 ± 55	81 ± 71	75 ± 33
c24–c26	43 ± 45	74 ± 48	73 ± 95	63 ± 52	53 ± 74	46 ± 42
c26–c28	39 ± 42	70 ± 46	62 ± 65	58 ± 42	39 ± 40	29 ± 15
c28–c30	35 ± 45	66 ± 31	53 ± 51	51 ± 39	36 ± 26	25 ± 10
c30–c32	27 ± 49	58 ± 26	43 ± 48	44 ± 42	11 ± 21	8 ± 7
c32–c34	32 ± 44	63 ± 23	49 ± 43	52 ± 51	15 ± 19	1 ± 10
c34–c36	32 ± 40	61 ± 28	49 ± 44	54 ± 51	18 ± 23	9 ± 14
c36–c38	14 ± 45	57 ± 43	42 ± 51	45 ± 61	0 ± 29	0 ± 20
c38–c40	4 ± 66	58 ± 72	28 ± 76	38 ± 97	0 ± 43	0 ± 47

* Indicates Statistical Significance compared to the Control.

**Table 2 microorganisms-13-00848-t002:** Heavy metal analysis of plants (aerial and roots) and soil. BCF: Bioconcentration factor, TF: translocation factor. M: mustard, RG1: Abergain (ryegrass), RG2: Abergreen (ryegrass), RGWC: ryegrass grass and white clover, Ch: chicory.

**Treatment**	**Shoots Pb (mg/kg)**	**Roots Pb (mg/kg)**	**Total Plant Pb (mg/kg)**	**Soil Pb (mg/kg)**	**BCF**	**TF**
M	0.14 ± 0.11	0.42 ± 0.25	0.28 ± 0.18	4.4 ± 0.46	0.07 ± 0.04	0.35 ± 0.16
RG1	0.26 ± 0.12	0.6 ± 0.23	0.43 ± 0.16	5.0 ± 0.24	0.08 ± 0.02	0.47 ± 0.21
RG2	0.31 ± 0.02	0.94 ± 0.39	0.69 ± 0.21	4.5 ± 0.74	0.16 ± 0.04	0.41 ± 0.08
RGWC	0.41 ± 0.03	0.92 ± 0.36	0.66 ± 0.19	4.45 ± 0.30	0.15 ± 0.05	0.51 ± 0.17
Ch	0.33 ± 0.03	0.89 ± 0.50	0.61 ± 0.26	4.65 ± 1.33	0.13 ± 0.02	0.45 ± 0.16
**Treatment**	**Shoots Cu (mg/kg)**	**Roots Cu (mg/kg)**	**Total Plant Cu (mg/kg)**	**Soil Cu (mg/kg)**	**BCF**	**TF**
M	0.34 ± 0.09	0.36 ± 0.15	0.35 ± 0.07	0.11 ± 0.02	3.34 ± 1.05	1.27 ± 0.97
RG1	0.38 ± 0.13	0.5 ± 0.08	0.44 ± 0.03	0.18 ± 0.01	2.47 ± 0.32	0.83 ± 0.48
RG2	0.36 ± 0.16	0.71 ± 0.32	0.55 ± 0.18	0.14 ± 0.05	4.62 ± 1.81	0.46 ± 0.27
RGWC	0.3 ± 0.13	0.82 ± 0.42	0.56 ± 0.23	0.23 ± 0.07	2.64 ± 1.47	0.45 ± 0.28
Ch	0.23 ± 0.03	0.2 ± 0.02	0.21 ± 0.02	0.3 ± 0.08	0.80 ± 0.27	1.2 ± 0.24
**Treatment**	**Shoots Zn (mg/kg)**	**Roots Zn (mg/kg)**	**Total Plant Zn (mg/kg)**	**Soil Zn (mg/kg)**	**BCF**	**TF**
M	6.57 ± 1.25	3.96 ± 0.62	5.26 ± 0.91	8.94 ± 1.28	0.60 ± 0.15	1.66 ± 0.13
RG1	3.92 ± 0.56	4.95 ± 0.43	4.44 ± 0.24	8.77 ± 0.28	0.51 ± 0.03	0.80 ± 0.18
RG2	3.17 ± 0.31	5.29 ± 0.86	4.18 ± 0.45	7.9 ± 0.89	0.54 ± 0.07	0.53 ± 0.09
RGWC	2.97 ± 0.46	5.55 ± 0.90	4.26 ± 0.44	7.95 ± 0.89	0.54 ± 0.10	0.55 ± 0.13
Ch	3.29 ± 0.33	3.69 ± 0.63	3.49 ± 0.38	8.03 ± 0.84	0.43 ± 0.03	0.91 ± 0.15

## Data Availability

The data presented in this study are openly available in [NCBI] at [https://www.ncbi.nlm.nih.gov, accessed on 15 September 2024], reference number [PRJNA1054554 and PRJNA1055331].

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
