# Peer review of "Assessing Microbial Activity and Rhizoremediation in Hydrocarbon and Heavy Metal-Impacted Soil"

_microorganisms, 2025, doi:10.3390/microorganisms13040848_

Round 1
Reviewer 1 Report
Comments and Suggestions for Authors
The manuscript microorganisms-3490519 describes the results of phytoremediation by several plants, presenting data on changes in soil microbial communities. The authors presented a large amount of experimental data. However, in my opinion, the manuscript contains many mistakes, and not all the arguments of the authors are properly substantiated.
Major remarks:
1. The manuscript uses erroneous/outdated names of many bacterial taxa, such as Proteobacteria, Firmicutes, Actinobacteriota. In addition, Saccharimonadales is used by the authors as a genus, but this group is an order. The scientific names of the plants (lines 116-118) are also incorrectly stated.
2. Many statements and assertions of the authors in the Introduction and Discussion sections are not supported by references or properly agreed upon. Overall, for a 25-page manuscript, the number of references could be significantly higher than 46.
3. The manuscript presents experiments on soil contaminated with hydrocarbons and heavy metals. And this is correctly reflected in the title of the manuscript. However, in the Abstract, Introduction, and Discussion, the authors focus only on hydrocarbons and microbes capable of degrading them. The authors do not discuss the impact of heavy metals on microbial communities.
Minor remarks:
1. Line 99: explain the meaning of "lime content 35.08%"? This value is higher than 29.8% silt (line 95).
2. Line 101: In the Materials and Methods section, it is indicated that the hydrocarbon content in the soil was 4 g/kg, but in Figure 2, this value is higher than 7 g/kg. What is the reason for such a difference in PHC values?
3. Lines 216-217: Provide formulas or references for calculating BCF and TF.
4. For what purpose was the comparison and statistical analysis of the biomass of different plants carried out?
5. Lines 246-248: mustard biomass does not differ significantly from the biomass of other plants.
6. Table 1: How should “0 ± 56”, “0 ± 90”, “0 ± 46” and other similar values ​​in the table be interpreted?
7. Table 2: Why was the concentration of metals in the plant obtained by adding up the concentrations in the shoots and roots? How is the concentration of metal in the whole plant higher than the concentration of metal in both the shoots and roots??? With equal masses of shoots and roots, the concentration in the plant should be close to the average value, not the sum.
8. Section 3.4: When listing phyla, families, and genera, use order from most abundant to the least abundant groups.
9. The authors do not discuss in any way the significant increase in the representation of Stenotrophomonas and Achromobacter.
10. Lines 583-587: Provide references showing the association of listed bacteria with the degradation of PHCs.
Author Response
Major remarks:
1. The manuscript uses erroneous/outdated names of many bacterial taxa, such as Proteobacteria, Firmicutes, Actinobacteriota. In addition, Saccharimonadales is used by the authors as a genus, but this group is an order. The scientific names of the plants (lines 116-118) are also incorrectly stated.
Though names may be outdated we were limited by what is contained within the SILVA database which was used to for bioinformatics. It is not possible for us to alter this at this stage. I have added an explanation to clarify this on lines 194-195.
Saccharimonadales appears as a genus due it being an unclassified species in the SILVA database. We have amended any instances of this being mentioned in references to genus as “unclassified Saccharimonadales sp.” See below to show how this ASVs taxonomy was contained within the SILVA ASV table:
ASV24 k__Bacteria;p__Patescibacteria;c__Saccharimonadia;o__Saccharimonadales;f__Saccharimonadales;g__Saccharimonadales
Regarding the incorrect scientific names for plants, we have reviewed all instances of binomial names and can confirm that Sinapis alba for mustard, Cichorium intybus for chicory, Lolium perenne for ryegrass, and Trifolium repens for white clover are the correct scientific names. If you are referring to “Abergain and Abergreen” these are the varieties of rye grass used, which was included as I stated directly before that two Lolium perenne varieties were used I felt it necessary to name and differentiate them.
- Many statements and assertions of the authors in the Introduction and Discussion sections are not supported by references or properly agreed upon. Overall, for a 25-page manuscript, the number of references could be significantly higher than 46.
Some references were unintentionally omitted from the introduction relating to use previous use of mustard, ryegrass and chicory for phytoremediation and phytoextraction respectively. There were in the reference list and used in the conclusion but accidently left out of the introduction. Something similar happened with the discussion where a reference was used only once when it was referred to in several other paragraphs and this has also been amended. Additionally, the number of references has been increased to 58 to address some statements that weren’t sufficiently supported. The entire discussion was restricted for clarity. Lines 559-730.
- The manuscript presents experiments on soil contaminated with hydrocarbons and heavy metals. And this is correctly reflected in the title of the manuscript. However, in the Abstract, Introduction, and Discussion, the authors focus only on hydrocarbons and microbes capable of degrading them. The authors do not discuss the impact of heavy metals on microbial communities.
There was little emphasis on heavy metals in this experiment by design. The priority was hydrocarbon degradation, but as the soil was co-contaminated phytoextraction was explored also. If you refer to table 2, specifically the section referring to concentration of metals in soil, you will see that there is less than 1 mg/kg difference (sometimes much less) between and within treatments at the end of the trial. For this reason, it’s not really possible to discuss the impacts of metals on the microbial communities as metal concentrations are uniform throughout every replicate and treatment. We agree that it would be more complete to assess this but it was never within the scope of the experiment. We have added information on metals results to the abstract also lines 21-23.
Minor remarks:
1. Line 99: explain the meaning of "lime content 35.08%"? This value is higher than 29.8% silt (line 95).
The lime content of 35.08% refers to the proportion of calcium carbonate present in the soil, which is distinct from the soil's textural components (sand, silt, and clay). The texture of the soil is described as sandy-loam, with the particle size distribution being 58.3% sand, 29.8% silt, and 11.8% clay. The lime content is separate from the soil's particle size fractions and does not contribute to the sum of sand, silt, and clay. Therefore, the lime content is reported as a separate parameter that influences soil pH and other chemical properties, while the silt, sand, and clay percentages are part of the textural classification. The total composition of the soil, including both mineral and organic components, does not exceed 100% when these factors are appropriately differentiated.
- Line 101: In the Materials and Methods section, it is indicated that the hydrocarbon content in the soil was 4 g/kg, but in Figure 2, this value is higher than 7 g/kg. What is the reason for such a difference in PHC values?
This was explained on lines 237-245 where we state: “It is very important to note that the soil tested contains a mixture of aromatics and aliphatics, however only alkanes were used to formulate the calibration curves. Any aromatics extrapolated on these curves would have a stronger signal than equivalent alkanes (due to unsaturation and π electrons). As a result, the total petroleum hydrocarbon concentration determined by GCFID in this study is higher than that referenced in the soil conditions section due to a significant number of aromatics present in the soil. However, the primary goal is to assess the relative changes between treatments, so even though the absolute values might be inflated, the proportional difference should remain consistent and therefore meaningful.”
- Lines 216-217: Provide formulas or references for calculating BCF and TF.
We have added the formulae to line 256. - For what purpose was the comparison and statistical analysis of the biomass of different plants carried out?
Given past links of plant biomass to hydrocarbon degradation and the importance of roots as a substrate for rhizospheric organisms we felt that would be useful to include these comparisons.
- Lines 246-248: mustard biomass does not differ significantly from the biomass of other plants.
Yes in terms of total biomass there was no statistically significant difference, this was stated in error. The significant difference was between mustard root biomass and RG2 root biomass. The line has been amend now to refer the reader to the contents of Table S5 which contains a table of average biomass for aerial root and total biomass + standard deviations (now lines 292-293).
- Table 1: How should “0 ± 56”, “0 ± 90”, “0 ± 46” and other similar values ​​in the table be interpreted?
This table refers to % degradation therefore if the average value is higher than the initial hydrocarbon concentration the degradation is considered to be zero. However, there is still a deviation between replicates so some replicates have degradation while others are higher than the initial level. This is more likely to happen with the low molecular weight hydrocarbons as the initial concentration was so small to begin with.
- Table 2: Why was the concentration of metals in the plant obtained by adding up the concentrations in the shoots and roots? How is the concentration of metal in the whole plant higher than the concentration of metal in both the shoots and roots??? With equal masses of shoots and roots, the concentration in the plant should be close to the average value, not the sum.
This was due to a calculation error in the raw data. Thank you for noticing this, it was not picked up by any other reviewers and certainly impacted results. We have recalculated the data, amended the results section lines 374-386, the and discussion line 731-748.
- Section 3.4: When listing phyla, families, and genera, use order from most abundant to the least abundant groups.
Thank you, this has been amended now lines 455-475
- The authors do not discuss in any way the significant increase in the representation of Stenotrophomonas and Achromobacter.
We have added several lines and 4 references to support any statements made on lines 697-706
- Lines 583-587: Provide references showing the association of listed bacteria with the degradation of PHCs
These citations have been added to line 673.
.
Reviewer 2 Report
Comments and Suggestions for Authors
Ms. 'Assessing Microbial Activity and Rhizoremediation in Hydrocarbon and Heavy Metal Impacted Soil' by Conlon et al. is well written and has an interesting topic.
Introduction engages the reader, setting the scene for the proposed research with a clear focus, purpose and direction on a relevant topic. The aim is also well highlighted. However, my concern is related to previous studies. They are not sufficiently well highlighted.
Materials and Methods: discuss and explain in detail the data collection and analysis methods you used in your research.
Results: relevant reports obtained concisely and objectively, in a logical order.
Keep plant species names consistent.
Discussion: The authors elaborate on the meaning, importance and relevance of the results. Unfortunately, there is no flow to this section, from discussing one result it turns abruptly to another. Also, references are very few.
Conclusion: clear and succinct.
Again, it is 'Mustard', and 'S. alba'. Elsewhere there is also 'mustard'.
Author Response
Introduction engages the reader, setting the scene for the proposed research with a clear focus, purpose and direction on a relevant topic. The aim is also well highlighted. However, my concern is related to previous studies. They are not sufficiently well highlighted.
Additional references and text have been added to lines 86-94 to highlight prior work on Sinapis alba, Lolium perenne, Cichicorium intybus and their roles in hydrocarbon and/or heavy metals remediation.
Materials and Methods: discuss and explain in detail the data collection and analysis methods you used in your research.
We have made several adjustments to the materials and methods section, including adding further information on soil conditions (lines 113-119), added a table to supplementary materials that explains details of the pot trial more clearly, re-wrote microbiome sampling section (lines 153-179), added more to GCFID sample preparation (lines 206-219) and re-writing the statistics (lines 252-263).
Results: relevant reports obtained concisely and objectively, in a logical order.
We appreciate the reviewers positive feedback on how we assembled and organised our results.
Keep plant species names consistent.
We added lines 97-101 that establish both the binomial and common names of the plants use and clarify that the common names will be used going forward in the text. All references to binomial names after this point have been amended.
Discussion: The authors elaborate on the meaning, importance and relevance of the results. Unfortunately, there is no flow to this section, from discussing one result it turns abruptly to another. Also, references are very few.
We have completely revised and restructured discussion lines 557-728. Hopefully the flow is improved, I tried to address each treatment type one at a time. However, I kept comments on fungal community and metals at the end.
Conclusion: clear and succinct.
Again, the positive feedback is much appreciated, thank you.
Again, it is 'Mustard', and 'S. alba'. Elsewhere there is also 'mustard'.
This has been addressed as described above.
Reviewer 3 Report
Comments and Suggestions for Authors
Review Report
This study presents a comprehensive and well-structured evaluation of rhizodegradation as a viable strategy for remediating petroleum hydrocarbon (PHC) and heavy metal co-contaminated soils. By employing a microcosm-scale pot trial, the research effectively examines the influence of different plant species on pollutant degradation and microbial community enrichment. The results highlight that Sinapis alba achieved the highest PHC degradation (68%) and fostered the enrichment of hydrocarbon-degrading bacterial taxa, underscoring its potential for phytoremediation applications. The manuscript is well-prepared, methodologically sound, and contributes valuable insights into plant–microbe interactions in contaminated soils. However, several aspects require further clarification and refinement to enhance the study’s rigor and broader applicability.
Comments for Authors
- The study mentions using six treatments (five plant treatments and one control), but it does not explain why these specific plant species were chosen in more detail beyond mentioning their root structures. Providing references or past studies supporting these choices would strengthen the justification.
- The Kruskal-Wallis test is used for PHC data, but the methods also state that the data was normally distributed—which contradicts the usual assumption for Kruskal-Wallis. If normally distributed, ANOVA may have been more appropriate.
- While statistical comparisons are made, they are inconsistent. For example:Chicory's root biomass was not statistically different from Ryegrass, but was statistically greater than mustard (P<0.05). This means mustard had significantly lower root biomass than chicory, but it is unclear if mustard was also significantly different from Ryegrass. The statement "no statistically significant difference between any of the Ryegrass treatments" suggests all ryegrass varieties performed equally, but this should be explicitly backed with p-values or confidence intervals. Bonferroni or FDR correction is not mentioned for multiple comparisons, which increases the risk of false positives in statistical tests.
- The control group showed large variability (3-81%) in PHC degradation. This wide range questions the reliability of comparisons, yet the study still compares mean degradation across treatments without addressing how this variability may affect statistical significance. The results say, "No treatment achieved better degradation than mustard by a statistically significant amount," yet mustard was only statistically different from some treatments, not all. The wording should be more precise.
- Spearman and Kendall’s correlation are both used, but the results are not directly compared. Kruskal-Wallis and Tukey’s test are used for diversity indices, but it is unclear why non-parametric tests were used in some cases and parametric ones in others. The data should be tested for normality consistently before applying these tests.
- Ryegrass had higher biomass with no statistical difference between treatments, yet its PHC degradation was not significantly higher than the control. This discrepancy is not explained—higher biomass might suggest greater microbial root interactions.
- The study states that "no link was found between plant biomass and PHC degradation" but does not offer hypotheses as to why. For instance: Was microbial activity more important than plant biomass? Did different root exudates influence degradation?
- The statement that hydrocarbon levels were at the threshold of plant growth inhibition lacks supporting citations or physiological evidence (e.g., root morphology changes, stress indicators).
Author Response
Comments for Authors
- The study mentions using six treatments (five plant treatments and one control), but it does not explain why these specific plant species were chosen in more detail beyond mentioning their root structures. Providing references or past studies supporting these choices would strengthen the justification.
We have added references to address this (lines 129 – 133). This explanation was already present in the text but the references were not in the correct place, this has been amended. We also added reference to chicory use for phytoextraction with a reference that had not been included before.
- The Kruskal-Wallis test is used for PHC data, but the methods also state that the data was normally distributed—which contradicts the usual assumption for Kruskal-Wallis. If normally distributed, ANOVA may have been more appropriate.
This was an error in the text, one-way ANOVA was completed on normally distributed data (total petroleum hydrocarbons and biomass data). Kruskal-Wallis was also performed, but on non-normally distributed data (alpha diversity metrics and fractional hydrocarbon analysis.) Lines 243-254 have been re-written for improved clarity and correct errors.
- While statistical comparisons are made, they are inconsistent. For example:Chicory's root biomass was not statistically different from Ryegrass, but was statistically greater than mustard (P<0.05). This means mustard had significantly lower root biomass than chicory, but it is unclear if mustard was also significantly different from Ryegrass. The statement "no statistically significant difference between any of the Ryegrass treatments" suggests all ryegrass varieties performed equally, but this should be explicitly backed with p-values or confidence intervals. Bonferroni or FDR correction is not mentioned for multiple comparisons, which increases the risk of false positives in statistical tests.
We added improvements to results relating to aerial and root biomass have been address on lines 271-278 and 287-295 respectively. Reference to any differences that were not statistically significant have been removed. Clarification on the use of Bonferroni corrections was added to 249-250.
- The control group showed large variability (3-81%) in PHC degradation. This wide range questions the reliability of comparisons, yet the study still compares mean degradation across treatments without addressing how this variability may affect statistical significance. The results say, "No treatment achieved better degradation than mustard by a statistically significant amount," yet mustard was only statistically different from some treatments, not all. The wording should be more precise.
Regarding the control variability, we initially addressed it in lines 546-551, however, we have added addition information and supporting references to lines 551-556. We feel there are many factors which may contribute to this result. We removed the imprecise sentence relating to mustard statistical significance and rephrases on line 349-350.
- Spearman and Kendall’s correlation are both used, but the results are not directly compared. Kruskal-Wallis and Tukey’s test are used for diversity indices, but it is unclear why non-parametric tests were used in some cases and parametric ones in others. The data should be tested for normality consistently before applying these tests.
Line 302-316 were completely re-written with a focus on comparing Spearmans and Kendall’s results. Hopefully this has improved clarity. An explanation on the use of non-parametric tests was added to the re-written statistics section mentioned earlier on lines 243-254.
- Ryegrass had higher biomass with no statistical difference between treatments, yet its PHC degradation was not significantly higher than the control. This discrepancy is not explained—higher biomass might suggest greater microbial root interactions.
Lines 546-556 discussing the variation in control results may explains discrepancy for lack of statistical significance. 615-623 offers further discussion such as no correlation found between plant biomass and hydrocarbon concentration, and explanations relating to the potential presence of indigenous degraders.
- The study states that "no link was found between plant biomass and PHC degradation" but does not offer hypotheses as to why. For instance: Was microbial activity more important than plant biomass? Did different root exudates influence degradation?
This was partially discussed, citing the possibility of indigenous hydrocarbon degraders in the soil, but has been expanded to include reference to root exudates. See lines 615-623.
- The statement that hydrocarbon levels were at the threshold of plant growth inhibition lacks supporting citations or physiological evidence (e.g., root morphology changes, stress indicators).
After reviewing the document I can’t find reference to this claim. However, a statement was made to address another reviewer comment on lines 556-558 stating that no visible signs of phytotoxicity were observed.
Reviewer 4 Report
Comments and Suggestions for Authors
The authors have accumulated a mass of facts in the phytoremediation arena. However both rye and mustards are well known to be accumulators for heavy metals as reported in many published papers. Here additional contamination by PHs are added in.
The data are interpreted by speculation only as this paper is a presentation of facts.
Strengths are the comparisons between plantings and assessment at more than one time point during growth.
The work stimulated me to think and add several comments based on results.
There are many formating issues, also pointed out as sticky notes.
I am still unclear of the planting schemes and the degree of replication- these must be clarified
Do not understand the rationale for clover with the rye - clover was not added to the brassica or chicory neither of which are legumes either there are likely free N fixing bacteria at work
One thought to focus the paper is to drop the fungal work or relegate this to supplemental only: this would emphasize the bacterial findings better

Author Response
The authors have accumulated a mass of facts in the phytoremediation arena. However both rye and mustards are well known to be accumulators for heavy metals as reported in many published papers. Here additional contamination by PHs are added in.
We acknowledge that both rye and mustard are well known accumulators of heavy metals, we noticed that the supporting references for this were not placed in the experimental layout section(though were referenced later in the text) and have been added to lines 129-133. This should clarify the past research in this area. This research aims to add data on PHC and heavy metal co-contamination remediation.
The data are interpreted by speculation only as this paper is a presentation of facts.
Our interpretations are based on statistically significant trends observed in PHC degradation rates, microbial community shifts, and plant biomass data. Where speculation is made it is presented as a hypothesis supported by statistical analysis. We have taken care to not draw conclusions without basis, however, we would re-examine any specific issues upon request.
Strengths are the comparisons between plantings and assessment at more than one time point during growth.
We appreciate the reviewer’s recognition of the strengths of out study, particularly the use of multiple sampling times, which allowed for better tracking of the microbial community.
The work stimulated me to think and add several comments based on results.
We appreciate the reviewer’s engagement with our results and the opportunity to further refine our manuscript based on their comments.
There are many formating issues, also pointed out as sticky notes.
We have carefully addressed the formatting issues noted in the document. We also addressed the additional sticky notes which asked for several other important additions such as adding information on PHC effect on waterflow in the introduction, improving descriptions of sampling procedure and addition of tables to the supplementary section to clarify planting. Several comments asked for information on other soil properties like humic substances, unfortunately these are not available but we have mentioned this in the text.
I am still unclear of the planting schemes and the degree of replication- these must be clarified
We have made attempts to clarify the planting procedure and hopefully the new version is satisfactory. The requested table was also created and added to the supplementary files.
Do not understand the rationale for clover with the rye - clover was not added to the brassica or chicory neither of which are legumes either there are likely free N fixing bacteria at work
The combination of rye and white clover has been used in previous phytoremediation research, including field trials conducted as part of the broader research group to which we belong. One key aspect that would have been investigated in this study is whether there was a statistically significant difference in remediation between ryegrass with and without white clover. However, since white clover did not germinate, this variable was not examined.
Planting mustard and chicory with white clover was not initially considered. However, after reviewing relevant literature in light of this comment, I found that the nutrient uptake dynamics between mustard, chicory, and ryegrass differ significantly, meaning these plants would not typically benefit from co-planting with legumes in the same way.
One thought to focus the paper is to drop the fungal work or relegate this to supplemental only: this would emphasize the bacterial findings better
We acknowledge the reviewer’s suggestion and have moved the fungal community results to the supplementary materials.
Round 2
Reviewer 1 Report
Comments and Suggestions for Authors
Thanks to the authors for their detailed responses to my comments. The manuscript has become much better after the changes, but I still have a few comments.
1) In their response, the authors state that studying the effects of heavy metals was not part of their research goals. However, the authors are working with soil with complex contamination. And I find that in such a complex soil contamination, it is a mistake to try to explain the observed results only through the hydrocarbon degradation. The resistance of microorganisms and plants to heavy metals may play a key role in reducing the toxicity of the soil for hydrocarbon-degrading bacteria. In my opinion, this is what can explain the increase in the number of bacteria of the genera Achromobacter and Stenotrophomonas in the first half of the experiment.
2) When discussing the results, the authors primarily focus on the role of plants. At the same time, for an article in the journal Microorganisms, the main focus should be on bacteria and fungi.
Minor remarks:
The authors should double-check the values presented in Table 2. Some values appear to be clearly erroneous (e.g., M – Total Plant Cu; RG1 – Roots Zn; and other) or inconsistent with the formulas in Section 2.6 (e.g., Ch – TF; M – TF; and other).
Line 827: Provide references that support the claim that Lysobacter, Rhodanobacter, and Massilia are associated with hydrocarbon degradation. References [36-38, 46 and 51] do not show this.
Line 868: Why is reference [57] used here? This reference is not related to Achromobacter.
Lines 887-904: I disagree with the authors: 1) The decrease in lead and zinc concentrations in soil is greater than the decrease in copper concentrations; 2) for copper, RG2 shows a reduction in copper concentration in soil equal to that of mustard, with a higher BCF value, and Ch has a higher TF value than mustard.
Author Response
1) In their response, the authors state that studying the effects of heavy metals was not part of their research goals. However, the authors are working with soil with complex contamination. And I find that in such a complex soil contamination, it is a mistake to try to explain the observed results only through the hydrocarbon degradation. The resistance of microorganisms and plants to heavy metals may play a key role in reducing the toxicity of the soil for hydrocarbon-degrading bacteria. In my opinion, this is what can explain the increase in the number of bacteria of the genera Achromobacter and Stenotrophomonas in the first half of the experiment.
We have added additional information on both genera and this resistance to various metals especially those tested for in this manuscript, along with relevant references. We also added this as a potential reason for the initial increase of both genera at week 3. This can be found at lines 712-720.
2) When discussing the results, the authors primarily focus on the role of plants. At the same time, for an article in the journal Microorganisms, the main focus should be on bacteria and fungi.
We greatly appreciate the reviewer’s perspective on this matter. While we understand the suggestion, we believe that the current organisation of the discussion best reflects the key drivers in our experiment. Specifically, the plant treatments were the major factors influencing changes in the microbial community, and we feel it is logical to structure the discussion around each plant species as they were the primary points of variation between treatments. We have discussed the microbial community shifts in detail within the context of these plant treatments, which we believe provides a clear and coherent narrative. Additionally, the current structure aligns with the scope of MDPI Microorganisms. We hope this rationale addresses the reviewer’s concerns, and we remain open to any further suggestions.
Minor remarks:
The authors should double-check the values presented in Table 2. Some values appear to be clearly erroneous (e.g., M – Total Plant Cu; RG1 – Roots Zn; and other) or inconsistent with the formulas in Section 2.6 (e.g., Ch – TF; M – TF; and other).
These are all transcription errors, the data is correct in the excel file but the errors were made when typing into the final table ( I couldn’t paste because I was using 2 devices so I could utilise 2 monitors). I have fixed these and listed what was changed: For M – Total Plant Cu 0.349 was erroneously rounded up to 0.40 rather than 0.35 (now fixed) Standard deviation was correct. RG1 -roots zinc error: 4395, meant to be 4.95 ( . and 3 beside each other on number pad). RG2 zinc TF 0.63 should have been 0.53. RGWC zinc BCF std deviation 0.13 should have been 0.10. RG2 BCF standard deviation not correctly rounded 0.06 changed to 0.07. RG2 TF Copper 0.68 changed to 0.46, typing error. RG2 TF lead 0.41 should have been 0.32
Line 827: Provide references that support the claim that Lysobacter, Rhodanobacter, and Massilia are associated with hydrocarbon degradation. References [36-38, 46 and 51] do not show this.
These have been added now: Lysobacter [24] (reference already in text), Rhodanobacter [53] and Massilia [52]. I also added some clarification on lines 670-671 that should have been present to begin with that some are associated with enrichment and / or degradation in hydrocarbon soils.
Line 868: Why is reference [57] used here? This reference is not related to Achromobacter.
This was a numbering error as there were major changes to the ordering of references and additional references were also added. The numbers were meant to be [55,56] not [56,57]. The same error was also present for the Stenotrophomonas references which were added at the same time and were corrected from [54, 55] to [53,54]. Now however more references were added so the numbers have changed again but are associated with the correct papers.
Lines 887-904: I disagree with the authors: 1) The decrease in lead and zinc concentrations in soil is greater than the decrease in copper concentrations; 2) for copper, RG2 shows a reduction in copper concentration in soil equal to that of mustard, with a higher BCF value, and Ch has a higher TF value than mustard.
We have added some amendments to lines 413-424 to hopefully improve clarity on statistically significant differences between TFs and BCFs. Our results show that for Copper, Mustard is higher (1.27) than Chicory (1.20) but neither of these are statistically significantly different than any other TF for copper.
I can’t find anything on the lines you referenced relating to reduction in soil metal concentrations, but this wasn’t something that was analysed in this experiment. Only one timepoint was used for determining metals in the soil and only one column is placed in Table 2 relating to soil metal concentration, so it is not possible to comment on any reductions. The purpose of determining the content in the soil was so that BCF could be carlculated. I hope this clarifies what we were trying to achieved.
Reviewer 3 Report
Comments and Suggestions for Authors
The authors have addressed all my questions, and the quality of the current version of the manuscript has significantly improved compared to the initial submission. I have no further comments, and therefore, I recommend that the manuscript be accepted for publication.
Author Response
Thanks you for your review of our manuscript
Reviewer 4 Report
Comments and Suggestions for Authors
Thank you for your detailed revisions. The work is most interesting and it is important that you discuss all the data just not the successful results
there are a few more comments as sticky notes - yes your work makes me think thank you

Author Response
Thank you for reviewing our manuscript